# Knowledge and Attitude of Polish Dental Healthcare Professionals during the COVID-19 Pandemic

**DOI:** 10.3390/ijerph182212100

**Published:** 2021-11-18

**Authors:** Anna Turska-Szybka, Maria Prokopczyk, Piotr Winkielman, Dorota Olczak-Kowalczyk

**Affiliations:** 1Department of Pediatric Dentistry, Medical University of Warsaw, Binieckiego 6 Str., 02-097 Warsaw, Poland; anna.turska-szybka@wum.edu.pl; 2Student Scientific Society at the Department of Pediatric Dentistry, Medical University of Warsaw, Binieckiego 6 Str., 02-097 Warsaw, Poland; bernatowa@gmail.com; 3Department of Psychology, University of California San Diego, 9500 Gilman Drive, La Jolla, CA 92093, USA; piotr@ucsd.edu; 4Faculty of Psychology, SWPS University of Social Sciences and Humanities, Chodakowska 19/31, 03-815 Warsaw, Poland

**Keywords:** COVID-19, dentist, pandemic, SARS-CoV-2, awareness, personal protective equipment

## Abstract

Objectives: This study analyzed Polish dentists’ knowledge of the SARS-CoV-2 virus and the main problems in their work during the early phase of the pandemic. Methods: Dentists responded to an online anonymous survey consisting of 57 questions relating to socio-demographics, knowledge about COVID-19, and office procedures. The obtained data were analyzed using basic descriptive statistics, significance of dependencies and Chi square and Mann–Whitney tests; *p* < 0.05. Results: Ultimately, responses from 730 dentists were included. The mean age was 43.62 ± 11.57. Almost 3/4 of the respondents followed the information on COVID-19. A total of 95.5% had knowledge about COVID-19. Genetic testing was the basic test according to 69.2%. Further, 56.0% were concerned about the pandemic, and 23.6% were significantly anxious. In addition, 42.1% considered a risk of infection with the SARS-CoV-2 in the workplace as very high. A total of 84.0% admitted patients performing a triage and using personal protective equipment (PPE). Further, 44.5% planned to become vaccinated. Continuing the work during the pandemic was strongly correlated with age, sector, and location and duration of work. Conclusions: Most Polish dentists follow the information on the COVID-19 protocol and have sufficient knowledge about COVID-19. Dentists are concerned and anxious about the situation. The vast majority admitted patients during the pandemic and use PPE. Only almost half plan to be vaccinated.

## 1. Introduction

At the end of 2019, the first cases of COVID-19 were diagnosed in Wuhan, China [1,2]. SARS-CoV-2 belongs to the beta-coronavirus family and can attach to the human angiotensin converting enzyme (ACE-2), which is also expressed in the oral cavity, making the virus potentially infectious to dentists [3].

In Poland, the first case was confirmed on 4 March 2020, and the state of epidemic lockdown was declared on 15 March 2020, which also led to interruption or reduction of work in dental practices across the country [4]. Dental practices were open again for patients since 27 April 2020. In this period a relatively low number of COVID-19 cases were diagnosed in Poland, compared to other countries. Data on SARS-CoV-2 in relation to dental procedures was limited and specific actions were insufficiently supported by scientific papers. Many dental healthcare practices provided emergency treatment only, or even chose to close completely, due to COVID-19 infection risk. Dental care in Poland is provided mainly by private dental healthcare professionals. Patients pay dentists directly [5]. According to Public Opinion Research Center CBOS data, the vast majority (69%) of Poles used the services of a private dental clinic (hence only 31% of Poles used the services reimbursed by the National Health Fund, i.e., less than 1/3) [6].

Because dental healthcare professionals work in close contact with the patients and are potentially exposed to aerosols produced during routine dental treatments, they are at risk of COVID-19 and respiratory infections [1,7,8]. Every workday they face hazards of many infectious diseases from viruses and bacteria present in patient’s oral cavity, so called occupational infectious diseases. Dental professionals are at risk of exposure to blood and oral fluids and to infectious microorganisms due to aerosol-generating procedures (AGP). Beyond HBV, hepatitis C virus (HCV), and HIV, the recent risk for disease transmission in healthcare settings is posed by COVID-19. Cumulatively, these risks can generate anxiety. The COVID-19 pandemic has greatly increased dentists’ infection-related fear, confusion and anxiety [9,10,11,12,13]. Ahmed et al. [11] found out among dentists from 30 countries worldwide that psychological effects of the COVID 19 pandemic include fear of their family members’ infection, anxiety, fear and economic anxiety. Studies on the psychological effects of epidemics and pandemics, including the COVID-19 pandemic report symptoms such as fear, anxiety about future, family and economics, stress, depression, sleep disorders, and other mental implications and disorders [14,15].

These fears are justified, as the dental profession is one of the most vulnerable to the COVID-19 pandemic [2,16]. The Centers for Disease Control and Prevention (CDC), the World Health Organization (WHO), and the American Dental Association (ADA) have prepared guidance and special epidemiological state orders regarding the containment of the spread of COVID-19 for dental healthcare professionals. They recommend restricting or suspending certain practices and the use of appropriate personal protective equipment and protocols of conduct regarding the COVID-19 pandemic [2,3,7,8,17,18,19,20,21,22,23,24,25,26]. To facilitate coping with challenges in healthcare, a A special COVID-19 application has been developed in Poland to predict and reduce the consequences of the pandemic [27].

SARS-CoV-2 virus infection can be detected by RT-PCR (real time-PCR) by taking material for testing from the upper respiratory tract, e.g., nasopharyngeal swabs or swabs from the throat (back wall) and nasal mucosa simultaneously [1,28]. The specificity and sensitivity of the RT-PCR test is very high. However, the test may produce false positives due to contamination of swabs, especially in asymptomatic patients, and false negatives in those in the recovery phase [1,28]. Another additional method of detecting patient infections are rapid serological tests with the detection of IgG and IgM antibodies, which were used by less than 2% of respondents in the office. The detection rate in the first week of infection for the PCR test is 66.7%, and for the antibodies test is 38.3% [1]. The advantage of the serological test is the ease of use and time, as it can be ready in 10 min, making it particularly useful in the dental office and can reduce the risk of virus transmission from the patient to the staff. However, note that many tests can give a false negative result in the early incubation phase of the disease, which significantly lowers the sensitivity of the test.

The knowledge of the characteristics of COVID-19 and risk factors for infection as well as management and the use of personal protective equipment (PPE) are keys for adequate investment of funds and protection of dental healthcare professionals and patients against possible exposure to SARS-CoV-2 [29]. After a visit, it is recommended to thoroughly disinfect the dental unit and other surfaces in the office with a 0.1% sodium hypochlorite solution for one minute and sterilize the instruments each time after admitting the patient [17]. During the pandemic, the Polish Dental Association (abbrieviated as PTS in Polish) recommended only urgent and accelerated visits and postponing scheduled visits (i.e., planned implantological, prosthetic, orthodontic treatment) [17]. WHO recommended that routine health care of the oral cavity (oral health controls, teeth cleaning, prophylaxis) and treatments in an esthetic’s dentistry field should be postponed until sufficient reduction in the rate of incidences of COVID-19 [30]. Dentists should favor procedures which reduce the quantity of aerosol dispersed into the environment [8,25,30]. Villani et al. [2] emphasize the utility of the use of rubber dam: it significantly reduces the risk of cross-transmission, decreases the aerosol by 70% and protects from the fluids from oral cavity.

Another key element of the recommendation is the protection of medical personnel and the selection of appropriate PPE protecting the respiratory tract (face filter (FFP2/FFP3) masks, surgical masks), eyes (goggles, helmets), body (aprons, caps) and hands (gloves) [17,25,30]. A special protective device has been even developed to reduce aerosol dispersion during dental treatment in the COVID-19 pandemic era.

If appropriate disinfection techniques and preventive measures are not applied, the next major source of infection may be radiographic examinations [31,32]. It is recommended to perform extraoral X-ray examinations, which, unlike the intraoral ones, do not cause coughing or gagging [3]. During a pandemic, pantomographic examination and CBCT become the method of choice [20]. The WHO currently recommends admitting the patient with an intraoral imaging withdrawal, especially in areas with high rates of viral spread and COVID-19 [30]. When using intraoral X-ray, it is recommended to double-secure sensors to avoid cross-infection and perforation [3].

The aim of the present study was to investigate knowledge and attitude of Polish dental healthcare professionals during the relatively early stage of COVID-19 pandemic with particular emphasis on the main problems in their work. The authors expected the current study to (i) provide valuable comparisons of different procedures on managing an office in a pandemic situation; (ii) gather information about Polish dentists’ attitudes and approaches to oral health care and protective measures, and (iii) define strategies to facilitate organization of the dental care during the current and likely future situations, so as to better confront similar outbreaks.

## 2. Materials and Methods

The structured survey was created using the free-access questionnaire on Google Forms application (google.com/forms/about/, accessed on 10 June 2020). The link to the online survey was sent through an anonymous mailing list to all dentists registered in The Polish Dental Association and Polish Society of Pediatric Dentistry. The survey was sent to 2470 dental healthcare professionals and 730 of them completed it (response rate 30.0%). The questionnaire was sent only once. Replies were obtained voluntarily at the turn of May and June 2020 for 14 days, when survey was open. The questionnaires were anonymous to maintain the privacy and confidentiality of all information obtained from patients during the study.

The questionnaire was designed by the investigators (A.T.-S., M.P., D.O.-K.). It was based on the literature concerning COVID-19 pandemic, the previously used questionnaires regarding COVID and other viral infections in the dental field following the Stehr–Green scale and professional guidelines [8,11,17,22,26,33,34,35,36,37].

The survey consisted of 57 questions (42 single choice, 15 multiple choice). The first part of the questionnaire included personal data (age, gender, working area and status), issues related to the COVID-19 pandemic (sources of knowledge about the SARS-CoV-2 virus, questions checking knowledge about the virus, incubation time, tests detecting infection, symptoms and routes of infection, ways to prevent the pathogen’s transmission, self-perceived risk of infection). The next part contained questions about working conditions and personal protective equipment (PPE) adopted after the pandemic outbreak (the number of patients admitted, procedures performed, additional changes introduced in the office, preventive measures for medical staff and patients, taking X-rays). Another part of the survey assessed the practitioners’ main concerns of the risk of infection for themselves and patients and anxiety. Information about possible SARS-CoV-2 vaccine was added to survey.

A preliminary questionnaire was presented to a small group of dentists (*n* = 10) to clarify questionnaire items. The intraclass correlation coefficient (ICC) for the repetition of the test and the intra-rater reliability of each item were calculated. An ICC value of 0.80 or higher was considered satisfactory. Items with an ICC value below 0.80 were discussed by the authors and modified and adjusted according to the preliminary results.

All the respondents of the survey completed an informed consent question embedded on the first page of the questionnaire. Participants who answered “YES” were automatically included in the survey. The inclusion criteria were as follows: the consent to the processing of personal data of participants and a fully answered questionnaire. The exclusion criteria were no written consent and incompletely filled questionnaire.

The obtained data were stored on a password protected computer drive. In first step the data were analyzed using basic descriptive statistics (e.g., percent of participants). Next, the significance of the relationship between the answers (continuing work) and the socio-demographic data was assessed using the Chi square test. However, for age and length of work, the dependence on the socio-demographic data was evaluated using the Mann–Whitney test *p* < 0.05 was considered significant in all statistical analyses. The statistical analysis was performed using the Statistica 13.1 (Dell Inc., Round Rock, TX, USA) program. The study was approved by Bioethics Committee of Medical University of Warsaw (decision number AKBE101/2020).

## 3. Results

Ultimately, our study included a total of 730 dentists aged 24–78 years (mean age 43.62 ± 11.57). Years of dental practice ranged from 1–30 years. The socio-demographic data of the surveyed dentists are presented in Table 1.

Among the respondents, 41.5% had specializations, including general dentistry (58.1%), pediatric dentistry (17.3%), conservative dentistry with endodontics (16.3%), dental surgery (9.3%), dental prosthetics (6.6%), orthodontics (6%) and periodontology (3.7%).

Almost 3/4 of the respondents (72.5%) followed the information on the COVID-19 pandemic on an ongoing basis. The most common sources of knowledge were information published by the Polish Dental Association (PTS) (73.4%) and the Ministry of Health (73.3%), online medical portals (70.7%), the least frequently Facebook (34.9%), and the press and medical magazines (37.4%).

The vast majority of respondents (95.5%) correctly identified the virus incubation time of as 1–14 days and correctly identified the most important symptoms of infection: cough (91.6%), fever (96.4%), shortness of breath (86.6%). Only half of dentists (53.5%) correctly stated that the highest viral load was probably at an early stage of infection. According to 69.2%, the primary test to detect SARS-CoV-2 infection was genetic RT-PCR, and 75.8% would not recommend ibuprofen as a pain reliever for a patient during a pandemic. The most common routes of viral infection were the droplet route (98.8%), contact with an infected person (72.6%) and touch (35.1%). When asked about the false information about the duration of the SARS-CoV-2 virus on surfaces, the respondents most often indicated “on plastic-up to 12 h” (35.3%), “on a copper surface-up to 4 h” (22.9%) and “on medical records-up to 24 h” (22.3%). According to 89.5%, high air humidity was conducive to the survival of the SARS-CoV-2 virus, and 58.8% chose 40% ethyl alcohol as a solution without confirmed viral killing efficacy. Dentists pointed out that COVID-19 infection can be prevented by disinfecting hands with an alcohol-based preparation (98.6%), maintaining hygiene (97.8%) and covering the nose and mouth with a tissue when coughing and sneezing (97.4%).

In response to the question: “Are you concerned about the pandemic situation?” 56.0% answered “rather yes” and 23.6% “very much”. Further, 42.1% of dentists stated that the risk of SARS-CoV-2 infection in their workplace was very high. In the opinion of 83.2% of the dentists, their office was properly prepared to work during the pandemic and 84.0% of dentists declared that they admit patients during the pandemic. Doctors who did not practice during the pandemic most often justified their decision with the fear of infection of the family (71.8%), themselves (62.4%) and the high costs of protection against infection (59.0%). The sense of responsibility for patients, especially regular patients (80.1%) was the main reason why dentists saw patients. The characteristics of admitting patients and working in the office are presented in Table 2.

The most common indications for patient admission were pain that could not be relieved by painkillers (98.7%), periodontal abscess (96.7%), and dental injuries (93.3%). Procedures that were not performed or avoided were sand-blasting (94.9%), scaling (71.0%) and making dental crown (60.2%). Dentists admitted that they are primarily guided by epidemiological recommendations for dental offices published on the websites of the Ministry of Health (82.9%) and Polish Scientific Societies (70.8%). The differences in the approach to application of protective measures and PPEs are summarized in Table 3, Table 4 and Table 5.

A total of 5% of the respondents indicated that a patient reported to them during the quarantine period with suspicion of SARS-CoV-2 virus infection or infected, who were admitted only in 12 cases (2.0%). Further, 58.4% of dentists knew the reference units where a patient in the quarantine period, suspected of being infected or infected, could be admitted. Face filter (FFP2/FFP3) masks (68.7%) were the personal protective equipment most difficult to obtain on the market during the pandemic, followed by disposable gloves (42.9%).

Among the respondents, 10.1% did not perform X-rays in the office, while others performed RVG (68%) and pantomogram (49.6%). Approximately 11% of dentists referred patients to CBCT (cone beam computer tomography). In the opinion of 21.4%, the current situation justified more frequent antibiotics’ prescribing in order to minimize the patient’s visits to the office. Only 1.8% of dentists used rapid tests to detect IgG and IgM antibodies to SARS-CoV-2 on patients prior to the visit or referred them to such tests. In the autumn-winter season, 25.8% of respondents vaccinated against influenza, while 44.5% planned to become vaccinated against COVID-19 when the vaccine is available. Further, 19.6% of dentists would volunteer in a hospital during the COVID-19 pandemic.

A significance of dependencies was observed between becoming vaccinated against flu in the autumn–winter season and the willingness to vaccinate against COVID-2019 in the long term (*p* = 0.00001; Chi square test). Continuing work during the pandemic was strongly depended on age (*p* = 0.007049, Mann–Whitney test), the sector of work, place and length of work, and general health (respectively: *p* = 0.00140; *p* = 0.01790; *p* = 0.02976; *p* = 0.01061, Chi square test). The relationships between continuing work during a pandemic and selected parameters is presented in Table 6. We have found a significant association between continuing work during a pandemic and being concerned about the situation related to the COVID-19.

The relationships between age and length of work and selected parameters are presented in Table 7. We found a significant association between age, length of work and being concerned about the situation related to the COVID-19, and between length of work and keeping up to date with relevant information about COVID-19, and usage of a rubber dam during treatment (Figure 1).

Anxiety was not particularly related to a specific field of dentistry. Concerning specializations with aerosol-generating procedures (AGP) and non-aerosol-generating procedures, dental surgery vs. other specializations or orthodontics vs. other specializations were not significantly different (*p* = 0.65450 vs. *p* = 0.49893, respectively; Chi square test). A significant association was found between anxiety and type of practice: GDP (general dental practitioner) vs. specialists (*p* = 0.00012; Chi square test). Feeling anxious about the situation related to the COVID-19 pandemic was significantly higher among general dental practitioner then specialists (Figure 2).

## 4. Discussion and Additional Results

There are several additional discoveries revealed by our study. These concern questions of knowledge, behavior, and adjustments made by dentists in the early stage of the pandemic. Up-to-date and reliable knowledge about the SARS-CoV-2 coronavirus allow for proper preparation and treatment of potentially infected patients and reduces the risk of infection of medical personnel. Accordingly, the key questions in our study concerned whether dentists obtain knowledge and the latest news from reliable sources. We also tested whether dentists’ knowledge and their behavior match the best practices and consensus recommended at the time (relatively early stage of pandemic, in the summer of 2020). Finally, we assessed how dentists adjusted their practices and treatments in response to the pandemic. Our data revealed the following information.

Dentists both in our study, as in related studies, most commonly used information from the professional societies/associations, WHO, National Ministry of Health, but also from accounts in social media [38,39,40].

Some of the most common symptoms of SARS-CoV-2 infection are fever, cough, loss of smell and taste and shortness of breath [1,8,28,37]. In our and other surveys, the respondents correctly selected the basic symptoms of SARS-CoV-2 infection [38,41]. The other symptoms are most often muscle pain, fatigue and an abnormal chest CT scan, while headache, dizziness, sputum production, hemoptysis, abdominal pain, nausea, diarrhea and vomiting are less common symptoms of COVID-19 disease [3].

SARS-CoV-2 virus can be spread directly by coughing, sneezing, airborne droplets or contact transmission (with the mucous membranes of the mouth, nose and eyes), through saliva, and through infectious material through contact with surfaces in the immediate vicinity or objects used on or by infected people [7,8,18]. In the present study, the droplet route was most often indicated.

SARS-CoV-2 virus infection can be detected by RT-PCR (real time-PCR) or rapid serological tests [1,28]. A total of 69.2% of respondents had adequate knowledge about patient testing.

There are measures that effectively reduce the infectivity of coronavirus within a minute, e.g., 62–71% or 78–95% ethanol, 0.5% hydrogen peroxide or 0.1–0.21% sodium hypochlorite [42,43]. More than half of the surveyed dentists (58.8%) knew the properties of agents against the coronavirus. Sarfaraz et al. [44] concluded that the surveyed dentists from 23 different countries across the world lacked adequate knowledge on the basic aspects of disinfection protocols.

With the spread of the pandemic emerged concerns about the dubious safety of ibuprofen due to its role in increasing the level of ACE2 in the renin-angiotensin-aldosterone system and the potential increased risk of contracting COVID-19 and/or worsening the COVID-19 infection [45]. Among surveyed dentists in the summer of 2020, 75.8% would not recommend it as a pain reliever for a patient during a pandemic. Note, however, it is currently believed that there is no evidence to discourage ibuprofen use [45].

Our study found a reduction in the number of admitted patients during the first wave of the pandemic, which is consistent with other studies [9,39,46,47,48,49,50]. Earlier research on the Polish dentists conducted at the beginning of the pandemic showed that as many as 71.2% of them completely suspended their practice in response to the pandemic [9]. In the current study, conducted over a month after the described one, only 16.0% of dentists declared the same. Both groups of respondents often justified this decision with fear of infection of family [9], which was also strongly emphasized by other dentists [11,39,49,51]. This could be the reason why 3.6% of our respondents changed their place of residence during the pandemic, and also why American dentists decided to self-isolate [52]. Similarities were also noticed in the reasons why other dentists decided to practice during the pandemic: a sense of responsibility especially for regular patients and helping patients in pain [9]. Moreover, in the cited study, a significant decrease in the number of patients admitted per week was noticed [9]. Our own survey showed an increase in the number of dentists working in only one office after the outbreak of the pandemic (from 54.7% to 69.5%), which may also indicate a reduction in the number of patients and the workload of doctors.

Difficult access to a dentist during the COVID-19 pandemic, longer waiting times for an appointment, strict restrictions and reducing the risk of infection related to aerosol transmission of the virus may contribute to more frequent prescription of painkillers and antibiotics by dentists [53] and the development of teledendistry as a solution to admitting patients [54]. According to the 21.4% of the surveyed, the current situation justified the frequent prescription of antibiotics to minimize patient visits in the office. US researchers observed that, a significant number of patients (15–20%) had received antibiotics prior to their emergency visit, mainly prescribed by dentists through telemedicine or by emergency medicine physicians in hospitals [52].

Due to the nature of dental practice, there is an increased risk of cross-infection with SARS-CoV-2 between dentists and patients, which requires detailed and effective means for control and prevention of infection [18]. Further, 68.2% to 92% of dental healthcare practitioners expressed fear of contracting SARS-CoV-2 during work [12,51]. Also, nearly half (42.1%) of our respondents assessed risk of infection with SARS-CoV-2 in their work as very high similarly to other studies [44,55].

Tysiąc-Miśta and Dziedzic [9] showed a significant impact of the recommendations of the Dental Association and the Ministry of Health on dentists’ decisions about commencing work in the office during a pandemic. The Polish Dental Association (PTS) at the end of March 2020 published detailed recommendations on the infection control strategy and protocol for patient management in the office [17]. In the era of a pandemic, tele-medicine triage is recommended (conducting an epidemiological interview before visiting the office), as well as creating an airlock, enabling hands’ disinfection to each person entering the office, non-contact measurement of temperature and completing epidemiological declaration [2,17]. Almost all respondents indicated that an epidemiological interview should be performed. According to other research, telemedicine triage was used by 56.6% of the respondents [51], 80% [55] and even 95% [56]. In Italian clinics staff measured the body temperature of patients much less frequently (in 23.5% in the Cagetti et al. study [55], in 25.2% in the Izzetti et al. study [56]) than in our own study (89.7%)).

It is important to rinse the patient’s mouth with 1% solution of hydrogen peroxide, 0.2% solution of povidone iodine, or alcoholic solution of 0.2% chlorhexidine [17]. Compared to 93% of the dentists in this study, only 24% of Ahmed et al. [11] respondents asked the patient to rinse their mouth with antibacterial fluid before treatment. In own analysis, patients rinsed their mouths before surgery, as in Italian studies: most often with 0.12–0.2% chlorhexidine solution [55] or hydrogen peroxide solution [56]. In comparison, in Karayürek et al. study [41] 30.4% dentist did not use any antiseptic solutions before dental examination. Similarly, patients were often supplied with disposable surgical masks at the entrance to the office (30.5% in our study and 32.76% in Cagetti et al. study [55]), and disposable shoe covers (31.6% in our study, 42.5% in Izzeti et al. study [56]).

Nearly one in three respondents offered visits only in emergency cases (urgent dental care, UDC), 60.2% did not perform the dental crown, and the overwhelming majority would withdraw from sandblasting (94.9%) and scaling (71%). Significant decrease in the number of conservative procedures was observed [47]. Dentist also reported admitting urgent patients for the following reasons: post-injury, with an abscess and those whose pain does not respond to painkillers. Abscesses and acute surgical procedures were also one of the mostly approved treatments [41,47]. A total of 99.7% of Italian dentists stopped working in April 2020 due to pandemic or limited it to UDC, pulp inflammation, prosthesis decementation and abscess [56]. According to recommendations, to reduce contamination and patient’s saliva, medical treatment should be carried out with a rubber dam, which was only 32.5% of our respondents claimed to use, while in Italy the rate was 75.8% [56]. Further, 84.8% Italian dentists believed that a rubber dam may be useful in the future during dental treatment [51]. The WHO recommended that routine health care should be postponed until sufficient reduction incidences of COVID-19 [30]. Minimal invasive dentistry and conservative approaches, such as the atraumatic restorative technique, partial or selective removal of carious tissues with hand instruments or manual scaling of calculus, should therefore, be preferred, when possible [57].

The use of protective measures, such as face filter (FFP2/FFP3) masks, was confirmed by about half of the dentists in Switzerland and Liechtenstein [40]. The surveyed dentists used face filter (FFP2/FFP3) masks more often (85.5%) than Italian dentists: 37.4% [56], 54.84% [55], and 62.2% [51]. In contrast, in a global study, 84% of dentists from different countries supported the use of masks with N95 filters for routine dental procedures in the epidemic era [11]. Almost all respondents (90.7%) used disposable surgical gowns as did other dentists [41,51] and on the contrary to Cagetti et al. study [55]—21.09%. Nearly three quarters of dentists from Italy used surgical masks, such as dentists in our study [51,55]. The great majority of British periodontists found that the suggested measures could be effective and were concerned about working without aerosol-generating procedures (AGPs) [58].

Because they are constantly exposed to bloodborne or airborne infections, either via direct or indirect contact, dental health care professionals should be aware of safety protocols to avoid contagion to protect both them and their patients [29]. Before the COVID-19 outbreak, some oral healthcare providers have used face masks, barrier face coverings, or surgical masks, but they were in minority. Conventional protective measures may not be sufficient during the respiratory disease outbreaks. The COVID-19 pandemic has elevated the additional number of PPE needed. However, the need for protection against the high infectivity of SARS-CoV-2, and awareness of the benefits from using masks, motivated dentists to apply more precautions and use advanced PPE. Using double gloves, face shields, N95 masks, patient’s pre-op rinsing mouth, precise disinfection may contribute to greater protection against HBV, HCV, HIV, influenza virus and other pathogens. Although COVID-19 infection control has improved attitudes to oral health care and protective measures, it may help to better confront similar future outbreaks, as shown by examples comparing readiness to use safety protocols in countries with or without prior Middle East respiratory syndrome coronavirus (MERS-CoV) or severe acute respiratory syndrome coronavirus (SARS-CoV) pandemic. However, it would not eradicated fears and anxiety about this disease.

Among the respondents, 68% performed RVG and pantomogram (49.6%) in the office, while 11% referred patients to CBCT (cone beam computer tomography). Note that CBCT is not an alternative to intraoral imaging due to its lower resolution, artifacts, and a much higher radiation dose [59]. However, digital imaging and tele-advice are intended to reduce the risk of cross-contamination [32] and only 45.75% of Italian dentists believed that digital dentistry may be used more often after the SARS-CoV-2 virus pandemic [51].

In the context of the current spread of the SARS-CoV-2 coronavirus, the COVID-19 vaccine is a reliable solution for prevention and monitoring a pandemic [3]. At the time of our survey, nearly half of our respondents expressed interest in a vaccination, when it becomes available, compared to the Zigron et al. study, in which almost twice as many specialists (85%) declared it [60]. Own results were then confirmed by data published by the Supreme Medical Chamber in March 2021 concerning vaccinated Polish dentists (44.5% in the survey/40% in the official Chamber’s data) [61]. However, data from August 2021 indicated that over 93% of Polish dentists were vaccinated with two doses [62]. According to public polls, 50% of Poles and 30% of Americans do not plan to become vaccinated against COVID-19, even when the vaccine is available [63,64]. Actually, vaccinations reduced dental professionals’ level of anxiety and fear [65]. It also resulted in an increase in the number of interventional procedures and a reduction in the amount of PPE usage in patients. However, the use of PPE should remain the main concern regardless of the vaccination [65].

Dentists who suspended their practice during the pandemic rated the risk of contracting COVID-19 much higher and were less familiar with the PTS and Ministry of Health guidelines than dentists who continued their work [9]. In some studies, the relationship between continuing work and age and length of work, having children, place of residence and belonging to coronavirus infection risk group due to comorbidities was not statistically significant [9], in contrast to our own research. In the Sarfaraz et al. study [44], the level of knowledge showed a statistically significant difference with respect to the years of practice and the origin country of dentists. Other results indicated that education level and work sector were significantly associated with average level of knowledge, while there was no significant difference due to level of knowledge, gender, or years of practice [36]. In Sinjari et al.’s study [51], no significant association was found between age and the fear of coronavirus infection, and an inverse relationship was found between rubber dam use and age, with a significantly difference between people under 40 and over 40 years of age, similar to own study. Dentists’ age correlated with beliefs that digital technology can help in future emergencies, such as pandemic [51]. In the present study, concern about the situation related to the COVID-19 pandemic was significantly correlated with age, length of work and continuing work during the pandemic. Constant monitoring new guidelines, practicing the profession with the highest risk of SARS-CoV-2 may have an impact on the mental health of dentists. Being aware of the increased risk of virus infection and having more responsibilities increased the stress of working with dental office staff [52]. According to the research by Consolo et al. [12], the decline in the professional activity of dentists was accompanied by a feeling of concern, anxiety and fear. Anxiety, insomnia, depression, obsessive-compulsive symptoms and somatization are all well-known psychological hazards for health care workers during a pandemic [66]. Most of respondents admitted concerns about their professional future [12,41,58].

More than two-thirds of the general dental practitioners (78%) from 30 countries questioned were anxious and scared by the devastating effects of COVID-19 [11]. In the present study, feeling anxious was significantly higher among general dental practitioner then specialists. Anxiety was not particularly related to a specific field of dentistry with aerosol-generating procedures (AGP) and non-aerosol-generating procedures. According to Kamal [67] study, stress and anxiety during COVID-19 was associated with different dental procedures, both aerosol and non-aerosol generating. However, severe stress was associated with scaling, root canal treatment, complex feelings and had non-significant associations with orthodontic and pediatric treatment. Stress or anxiety were not significantly associated with orthodontic treatment, most probably due to the nature of orthodontic appointments. Associations between anxiety and simple fillings, simple extractions and pediatric procedures were statistically non-significant [67]. Dentists perceive the risk of contracting COVID-19 due to aerosol and non-aerosol procedures similarly [68]. In a global study by Ahmed et al. [11] most dentists showed anxiety and fear related to overwhelming COVID-19’ consequences. Factors protecting against psychological distress were age, clinical experience, keeping up to date with information and taking precautionary measures, while the risk factors include, among others, being a woman, and being at high risk of contracting the coronavirus [69,70]. The statement is reflected in the study by Al-Amad and Hussein [71]: the anxiety level was independent on age, length of professional work and professional category. According to Mahdee et al. [72], anxiety among dentists working in the hospital and clinic was statistically significantly higher than among those admitting only in the clinic.

COVID-19 pandemic also had negative economic effects on dental services and led to a decrease in revenues. For example, during the first wave of the pandemic and lockdown, most German dentists reported a reduction in workload in the office by more than half [73]. Almost three-quarters of Swiss dentists rated an income drop by 80–100% in the most unprofitable month [50]. Admitting each patient in Belgium was associated with additional costs (EUR 10–30) and time (10–30 min) [74]. Similarly, in Polish study, nearly 70% respondents reported an increase in time to admit a single patient [46]. Nearly half of practitioners from Czech Republic noticed a smaller number of patients despite open offices in the period of March–May 2020 [49], likewise, a number of dentists from the Central European study reduced their working hours during first lockdown [50]. Moreover, the perception of the pandemic as a financial hazard and economic transformations are one of the important factors in increasing the level of distress [1,4,75,76].

The participants in the survey were not randomly selected, which is an important limitation of this study. Due to the short duration of the survey and sending it only once without reminding about the answer, the response rate was quite low. This resulted in a smaller than expected sample size, even though the sample is in some ways representative for Polish dentists. The short time of data collection was due to the constantly changing situation in Poland over the summer during the COVID-19 pandemic. Another limitation is a cross-sectional nature of this study which can qualify conclusions regarding the real levels of awareness of dentists in Poland. In the present study, over-reporting and desirability bias are additional limitations. Unfortunately, a potential response bias is a recognized limitation in all such questionnaire surveys.

As mentioned, the present study concerns the pre-vaccine period. With the advancement of vaccination and the development of new variants (like Delta), the protocols and requirements for PPE during dental treatments have been changing. The Delta variant is more contagious, and spreads more easily than previous forms and it can be transmitted by fully vaccinated individuals [77]. Hence, it is paramount to be constantly aware of the risk of infection in dental office even when admitting asymptomatic patients.

## 5. Conclusions

The current study focused on the early stage (first wave) of the pandemic. Here are some of the key findings about Polish dentists who responded to our survey. Most of the respondents follow the information about the COVID-19 pandemic on an ongoing basis. The vast majority have knowledge about the symptoms of infection, the routes of spreading of the virus and methods for preventing COVID-19 infection. Half of the respondents are concerned about the pandemic situation, and every fourth is significantly anxious. During the pandemic, the respondents see patients whose pain was not reduced by painkillers, with abscesses and tooth injuries; without sandblasting, rarely performing scaling and permanent prosthetic restorations. Most respondents are taking preventive measures against the virus, both for patients and themselves. At the time of the study, almost half plan to be vaccinated against COVID-19. Continuing work, age, and duration of work, and continuing to work during the pandemic were all significantly related to being concerned about the COVID-19 situation. Age and length of work were also significantly related to usage of a rubber dam during treatment and keeping up to date with relevant information. Overall, this study gives insight into the reactions of dentists during the early phase of one of the most important public health challenges in recent years.

## Figures and Tables

**Figure 1 ijerph-18-12100-f001:**
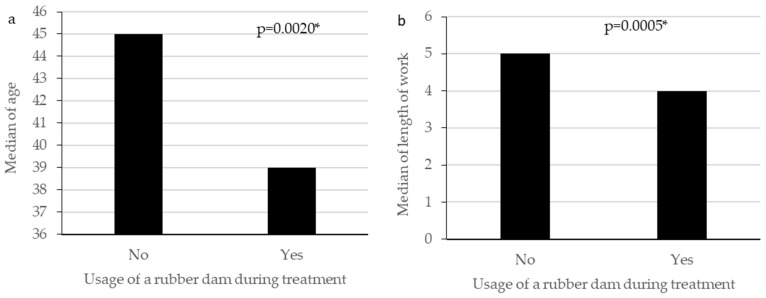
Relationships between usage of a rubber dam during treatment and age (**a**), length of work (**b**), the Mann–Whitney test (* statistically significant *p* < 0.005).

**Figure 2 ijerph-18-12100-f002:**
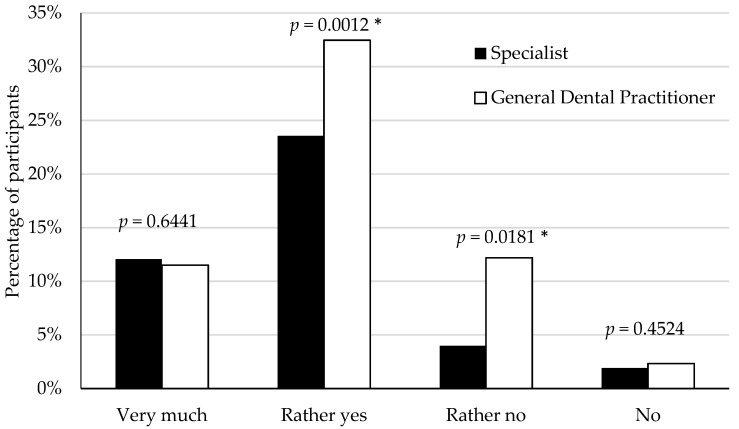
Feeling anxious about the situation related to the COVID-19 pandemic among general dental practitioner and specialists (* significant difference at *p* < 0.05 by way of the Chi square test).

**Table 1 ijerph-18-12100-t001:** Socio-demographic characteristic of participants.

Demographic Parameters		No. of Participants	% of Participants
Gender	Female	641	87.8%
	Male	89	12.2%
Work experience (in years)	1–5	119	16.3%
	5–10	120	16.4%
	10–20	150	20.5%
	20–30	225	30.8%
	>30	116	15.9%
Workplace	Big city	398	54.4%
	Small city	269	36.8%
	Village	63	8.6%
Type of activity	Private practice	372	51.0%
	Private practice and public structure (National Health Fund)	320	43.8%
	National Health Fund	38	5.2%
Number of units in workplace	1	294	40.3%
	2	203	27.8%
	3	90	12.3%
	4 or more	143	19.6%
Have own family/children and/or grandchildren	Yes	502	68.8%
	No	228	31.2%
Health—generally healthy	Yes	675	92.5%
	No	55	7.5%
Pre-pandemic number of workplaces	One	399	54.7%
	Two	239	32.7%
	Three or more	92	12.6%

**Table 2 ijerph-18-12100-t002:** Selected answers to questions about working in the office during a pandemic.

Category		No. of Participants	% of Participants
How many days a week do you see patients during a pandemic?	5 days	265	43.2%
	4	120	19.6%
	3	140	22.8%
	2	66	10.8%
	1	22	3.6%
How many hours a day do you see patients during a pandemic?	3–5 h	210	34.3%
	5–6 h	292	47.6%
	7–8 h and more	111	18.1%
How many workplaces do you work in after the outbreak of the pandemic?	1	426	69.5%
	2	153	25.0%
	3 and more	34	5.5%
How long does it take between admitting patients?	Directly after each other	31	5.1%
	15 min	123	20.1%
	Half an hour	133	21.7%
	An hour	326	53.2%
Does the patient additionally sign the consent related to treatment during the COVID-19 pandemic before the visit?	Yes	476	77.7%
	No	137	22.3%
Have you changed your place of residence to reduce the risk of infection of your family?	Yes	22	3.6%
	No	591	96.4%
What type of visits do you conduct?	Urgent dental care (UDC)	183	29.9%
	Scheduled visits	26	4.2%
	Emergency and urgent dental procedures and scheduled visits	404	65.9%
During the pandemic I treat:	Children	491	80.1%
	Adults	595	97.6%
	Elder people	394	64.3%

**Table 3 ijerph-18-12100-t003:** Protective measures for patients taken by dentists that continued to work after the outbreak of COVID-19.

Category		No. of Participants	% of Participants
Protective measures for patients	Epidemiological interview before the visit	599	97.7%
	Temperature measurement before the visit	550	89.7%
	Hand disinfection before entering the office	590	96.2%
	Rinsing the mouth with a disinfectant before the treatment	570	93.0%
	Treatment of the patient in a rubber dam	199	32.5%
Protective measures for the patient provided at the entrance to the office	Disposable surgical masks	187	30.5%
	Disposable gloves	292	47.6%
	Shoe covers	194	31.6%
	Disposable apron	168	27.4%
	Not applicable—I do not provide personal protective equipment	250	40.8%

**Table 4 ijerph-18-12100-t004:** Protective measures taken by dentists used for oneself and/or assistant.

Category		No. of Participants		% of Participants	
Protective measures used for oneself as a dentist and/or assistant		Dentist		Assistant	
	Face filter (FFP2/FFP3) masks	524	85.5%	402	65.6%
	Disposable surgical masks	467	76.2%	433	70.6%
	Disposable surgical aprons (protective outerwear, clothing)	556	90.7%	471	76.8%
	Disposable coveralls	276	45.0%	214	34.9%
	Shoe covers	329	53.7%	279	45.5%
	A disposable cap	531	86.6%	454	74.1%
	Protective goggles	272	44.4%	234	38.2%
	Protective glasses	351	57.2%	244	39.8%
	Protective visors/face shields	582	94.9%	497	81.1%

**Table 5 ijerph-18-12100-t005:** Protective measures used in the office taken by dentists that continued to work after the outbreak of COVID-19.

Category		No. of Participants		% of Participants	
Airlock in the office		For dentist		For patient	
	Yes	319	35.7%	135	22.0%
	No	394	64.3%	478	78.0%
Protective measures used in the office	Washing all surfaces after the visit with a disinfectant	566		90.7%	
	Spraying all surfaces with disinfectants after the visit	456		74.4%	
	Office ventilation, air exchange	578		94.3%	
	Office ozonation	109		17.8%	
	Disinfection with antibacterial lamps	404		65.9%	
	Washing protective fabric clothes in a washing machine at 60 °C for 40 min	484		79.0%	
	Usage of a biosanitizer	45		7.3%	

**Table 6 ijerph-18-12100-t006:** Relationships between continuing work during a pandemic and selected parameters—evaluated using the Chi square test.

	Keeping Up to Date with Relevant Information about COVID-19	Following the Guidelines of the Ministry of Health	Following the Guidelines of the Polish Dental Society	Are You Concerned about the Situation Related to the COVID-19?	Assessment of the Risk of Infection with SARS-CoV-2 Virus at Work
Continuing work during a pandemic	*p* = 0.0963	*p* = 0.3321	*p* = 0.0709	*p* = 0.0013 *	*p* = 0.0106

* Statistically significant (*p* < 0.005).

**Table 7 ijerph-18-12100-t007:** Relationships between age, length of work and selected parameters, evaluated using the Mann-Whitney test.

Category		Age		Length of Work	
		Median	*p* Value for the Mann-Whitney Test	Median	*p* Value for the Mann-Whitney Test
Keeping up to date with relevant information about COVID-19	(no)	35.40	*p* = 0.0123	3.00	*p* = 0.0002 *
	(occasionally)	34.00		3.00	
	(yes)	48.00		5.00	
Following the guidelines of the Ministry of Health	(no)	45.00	*p* = 0.1819	4.00	*p* = 0.6544
	(yes)	43.00		4.30	
Following the guidelines of the Polish Dental Society	(no)	40.00	*p* = 0.0167	4.50	*p* = 0.0892
	(yes)	45.00		4.00	
Are you concerned about the situation related to the COVID-19?	(no)	38.00	*p* = 0.0002 *	4.00	*p* = 0.0001 *
	(rather no)	36.00		3.00	
	(rather yes)	44.00		4.00	
	(very much)	49.00		5.00	
Assessment of the risk of infection with SARS-CoV-2 virus at work	(low risk)	41.00	*p* = 0.1202	4.00	*p* = 0.4901
	(medium risk)	41.00		4.00	
	(high risk)	43.00		4.00	
	(very high risk)	46.00		5.00	

* Statistically significant (*p* < 0.005).

## Data Availability

The datasets generated for this study are available on request to the corresponding author. After results have been published, an anonymized dataset will be made publicly available at an appropriate data archive.

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
