# Peer review of "Knowledge and Attitude of Polish Dental Healthcare Professionals during the COVID-19 Pandemic"

_ijerph, 2021, doi:10.3390/ijerph182212100_

Round 1

Reviewer 1 Report

First of all, I would like to congratulate the authors for this study, but I would like to suggest some minor improvements.

In "Introduction" and "Materials and Methods" sections, authors need do specify the length of time the research was conducted.

It's also important to include in the "discussion" section that, with the advancement of vaccination and the development of new variants (like Delta), the protocols and requirements for PPE during dental treatments have  been changing.

Author Response

Response to Reviewer 1 Comments

Dear Reviewer,

We would like to express our gratitude for your attention concerning our manuscript entitled "Knowledge and attitude of Polish dental healthcare professionals during the COVID-19 pandemic" and the useful comments and suggestions helping us to improve the paper. We truly appreciate the constructive criticisms of you. We acknowledge these realties
as limitations of our work. Yours efforts and insights were a tremendous help to us during this revision.

We have considered them carefully and made the required revision as addressed point-by-point below. We hope that you find these revisions an improvement and believe that
the revised version can meet your requirements.

We will appreciate the verification whether the present version of the manuscript fulfils the criteria and can be accepted as sufficiently modified. The authors will welcome further constructive comments if any.

Point 1: In "Introduction" and "Materials and Methods" sections, authors need do specify the length of time the research was conducted.

Response 1: Thank you for your comment. We added this information.

Revised text: The questionnaire was sent only once. Replies were obtained voluntarily at the turn of May and June 2020 for 14 days, when survey was open.

Point 2: It’s also important to include in the “discussion” section that, with the advancement of vaccination and the development of new variants (like Delta), the protocols and requirements for PPE during dental treatments have  been changing.

Response 2:  Thank you for your comment. We have included in the end of “discussion” section that, with the advancement of vaccination and the development of new variants (like Delta), the protocols and requirements for PPE during dental treatments have  been changing.

Revised text: As mentioned, the present study concerns the pre-vaccine period. With the advancement of vaccination and the development of new variants (like Delta), the protocols and requirements for PPE during dental treatments have been changing. The Delta variant is more contagious and spreads more easily than previous forms and it can be transmitted by fully vaccinated individuals [77]. Hence, it is paramount to be constantly aware of the risk of infection in dental office even when admitting asymptomatic patients.

Reviewer 2 Report

The topic of the manuscript is interesting in terms of evaluation of a particular group of health care professionals (dentist).

The following issues should be addressed:

Introduction: a paragraph regarding anxiety in dentist during everyday work, also for other possible risks, should be reported, to compare anxious sensations Covid-related with others related, e.g., to infective diseases

Methods: interval of time of the survey should be reported....Ethical approval was in 2020, so was the study conducted in 2020 or 2021?

Results: I suggest to add more tables to sum up some outcomes related to the text

Discussion: -consider to discuss if the attitude toward the others professional risks could have changed in positive or negative sense, considering that almost all of the protective measures are also useful in all cases (not only for Covid emergency)

-discuss if the anxiety sensations could be particularly related to specific field of dentistry..or if all fields were perceived in the same modality (e.g. surgery Vs non-invasive procedures with aerosol)

Author Response

Response to Reviewer 2 Comments

Dear Reviewer,

We would like to express our gratitude for your attention concerning our manuscript entitled "Knowledge and attitude of Polish dental healthcare professionals during the COVID-19 pandemic" and the useful comments and suggestions helping us to improve the paper. We truly appreciate the constructive criticisms of you. We acknowledge these realties as limitations of our work. Yours efforts and insights were a tremendous help to us during this revision.

We have considered them carefully and made the required revision as addressed point-by-point below. We hope that you find these revisions an improvement and believe that the revised version can meet your requirements.

We will appreciate the verification whether the present version of the manuscript fulfils the criteria and can be accepted as sufficiently modified. The authors will welcome further constructive comments if any.

Point 1: Introduction: a paragraph regarding anxiety in dentist during every day work, also for other possible risks, should be reported, to compare anxious sensations Covid-related with others related, e.g., to infective diseases

Response 1: Thank you for this valuable suggestion. We have added information about the dentist's fear caused by these factors.

Revised text: Because dental healthcare professionals work in close contact with the patients and are potentially exposed to aerosols produced during routine dental treatments, they are at risk of COVID-19 and respiratory infections [1,7,8]. During every work day they face hazards of many infectious diseases from viruses and bacteria present in patient’s oral cavity, so called occupational infectious diseases, which can result in anxiety. They are at risk of exposure to blood and oral fluids and to infectious microorganisms due to aerosol-generating procedures (AGP). Beyond HBV, hepatitis C virus (HCV), and HIV, the recent risk for disease transmission in healthcare settings is posed by COVID-19. The COVID-19 pandemic has led to an overwhelming feeling of fear, confusion and anxiety among dentists of being infected by their patients [9-13]. Ahmed et al. [11 ] found out among dentists from 30 countries worldwide that psychological effects of the COVID 19 pandemic include fear of their family members’ infection, anxiety, fear and economic anxiety. Studies on the psychological effects of epidemics and pandemics, including the COVID-19 pandemic, have reported symptoms such as fear, anxiety about future, family and economics, stress, depression, sleep disorders, and other mental implications and disorders [14,15].

Point 2: Methods: interval of time of the survey should be reported....Ethical approval was in 2020, so was the study conducted in 2020 or 2021?

Response 2: Thank you for your comment. We have specified the year and the exact duration of the study – May/June 2020, duration: 14 days.

Revised text: The questionnaire was sent only once. Replies were obtained voluntarily at the turn of May and June 2020 for 14 days, when survey was open.

Point 3: Results: I suggest to add more tables to sum up some outcomes related to the text

Response 3: Thank you for this valuable suggestion. We have added more tables and figures and modified previous table to sum up some outcomes related to the text.

The added tables, figures and modified previous table are as follows:

Table 3. Protective measures for patients taken by dentists that continued to work after
the outbreak of COVID-19.

Category

No. of

participants

% of

participants

Protective measures for patients

Epidemiological interview before the visit

599

97.7%

Temperature measurement before the visit

550

89.7%

Hand disinfection before entering the office

590

96.2%

Rinsing the mouth with a disinfectant before the treatment

570

93.0%

Treatment of the patient in a rubber dam

199

32.5%

Protective measures for the patient provided at the entrance to the office

Disposable surgical masks

187

30.5%

Disposable gloves

292

47.6%

Shoe covers

194

31.6%

Disposable apron

168

27.4%

Not applicable - I do not provide personal protective equipment

250

40.8%

Table 4. Protective measures taken by dentists used for oneself and / or assistant.

Category

No. of

participants

% of

participants

Protective measures used for oneself as a dentist and / or assistant

Dentist

Assistant

Face filter (FFP2/FFP3) masks

524

85.5%

402

65.6%

Disposable surgical masks

467

76.2%

433

70.6%

Disposable surgical aprons (protective outerwear, clothing)

556

90.7%

471

76.8%

Disposable coveralls

276

45.0%

214

34.9%

Shoe covers

329

53.7%

279

45.5%

A disposable cap

531

86.6%

454

74.1%

Protective goggles

272

44.4%

234

38.2%

Protective glasses

351

57.2%

244

39.8%

Protective visors/face shields

582

94.9%

497

81.1%

Table 5. Protective measures used in the office taken by dentists that continued to work after the outbreak of COVID-19.

Category

No. of

participants

% of

participants

Airlock in the office

For dentist

For patient

Yes

319

35.7%

135

22.0%

No

394

64.3%

478

78.0%

Protective measures used in the office

Washing all surfaces after the visit with a disinfectant

566

90.7%

Spraying all surfaces with disinfectants after the visit

456

74.4%

Office ventilation, air exchange

578

94.3%

Office ozonation

109

17.8%

Disinfection with antibacterial lamps

404

65.9%

Washing protective fabric clothes in a washing machine at 60oC for 40 minutes

484

79.0%

Usage of a biosanitizer

45

7.3%

Table 6. Relationships between continuing work during a pandemic and selected parameters, evaluated using the chi square test.

Keeping up to date with relevant information about COVID-19

Following the guidelines of the Ministry of Health

Following the guidelines of the Polish Dental Society

Are you concerned about the situation related to the COVID-19?

Assessment of the risk of infection with SARS-Cov-2 virus at work

Continuing work during a pandemic

p=0.0963

p=0.3321

p=0.0709

p=0.0013*

p=0.0106

*Statistically significant (p<0.005)

Table 7. Relationships between age, length of work and selected parameters, evaluated using the Mann-Whitney test.

Category

Age

Length of work

Median

p value for the Mann-Whitney test

Median

p value for the Mann-Whitney test

Keeping up to date with relevant information about COVID-19

(no)

35.40

p=0.0123

3.00

p=0.0002*

(occasionally)

34.00

3.00

(yes)

48.00

5.00

Following the guidelines of the Ministry of Health

(no)

45.00

p=0.1819

4.00

p=0.6544

(yes)

43.00

4.30

Following the guidelines of the Polish Dental Society

(no)

40.00

p=0.0167

4.50

p=0.0892

(yes)

45.00

4.00

Are you concerned about the situation related to the COVID-19?

(no)

38.00

p=0.0002*

4.00

p=0.0001*

(rather no)

36.00

3.00

(rather yes)

44.00

4.00

(very much)

49.00

5.00

Assessment of the risk of infection with SARS-Cov-2 virus at work

(low risk)

41.00

p=0.1202

4.00

p=0.4901

(medium risk)

41.00

4.00

(high risk)

43.00

4.00

(very high risk)

46.00

5.00

(yes)

39.00

4.00

*Statistically significant (p<0.005)

Figure 1. Relationships between usage of a rubber dam during treatment and age (a), length of work (b), the Mann-Whitney test (* statistically significant p<0.005).

Figure 2. Feeling anxious about the situation related to the COVID-19 pandemic among general dental practitioner and specialists (*significant difference at p<0.05 by Chi square test)

Point 4: Discussion: - consider to discuss if the attitude toward the others professional risks could have changed in positive or negative sense, considering that almost all of the protective measures are also useful in all cases (not only for Covid emergency)

Response 4: Thank you for this valuable suggestion. We have discuss the attitude toward the others professional risks.

Revised text: Because dentists are constantly exposed to bloodborne or airborne infections, either via direct or indirect contact, dental health care professionals should be aware of safety protocols to avoid contagion to protect both them and their patients [29]. Before the COVID-19 outbreak, some oral healthcare providers have used face masks, barrier face coverings, or surgical masks, but they were in minority. Conventional protective measures may not be sufficient during the respiratory disease outbreaks. The COVID-19 pandemic has elevated the additional number of PPE needed. However, forced by the need for protection against the high infectivity of SARS-Cov-2, and being aware of benefits from using such masks, dentists started to apply more precautions and use advanced PPE. Using double gloves, face shields, N95 masks, patient’s pre-op rinsing mouth, precise disinfection may contribute to greater protection against HBV, HCV, HIV, influenza virus and other pathogens. Although COVID-19 infection control has improved attitudes to oral health care and protective measures, it may help to better confront similar future outbreaks, as shown by examples comparing readiness to use safety protocols in countries with or without prior Middle East respiratory syndrome coronavirus (MERS-CoV) or severe acute respiratory syndrome coronavirus (SARS-CoV) pandemic

Point 5: - discuss if the anxiety sensations could be particularly related to specific field of dentistry..or if all fields were perceived in the same modality (e.g. surgery Vs non-invasive procedures with aerosol)

Response 5: Thank you for this valuable suggestion.

We have discussed association between anxiety and relation to all fields of dentistry.

We have also added in the Section “Results” a Figure 2. Feeling anxious about the situation related to the COVID-19 pandemic) among general dental practitioner and specialists (*significant difference at p<0.05 by Chi square test)” is as follows:

Revised text: More than two-thirds of the general dental practitioners (78%) from 30 countries questioned were anxious and scared by the devastating effects of COVID-19 [11 ] In the present study, feeling anxious was significantly higher among general dental practitioner then specialists. Anxiety was not particularly related to a specific field of dentistry with aerosol-generating procedures (AGP) and non-aerosol-generating procedures. According to Kamal [67] study, stress and anxiety during COVID-19 was associated with different dental procedures, both aerosol and non-aerosol generating. However, a severe stress was significantly associated with scaling, root canal treatment, complex feelings and had non-significant associations with orthodontic and pediatric treatment. Stress or anxiety were not significantly associated with orthodontic treatment, most probably due to the nature of orthodontic appointments. Associations between anxiety and simple fillings, simple extractions and pediatric procedures were statistically non-significant [67]. Dentists perceive the COVID-19’ contracting risk in similar extent concerning the aerosol and non-aerosol procedures [68]. In global study by Ahmed et al. [11] most dentists showed anxiety and fear related to overwhelming COVID-19’ consequences. Factors protecting against psychological distress were age, clinical experience, keeping up to date with information and taking precautionary measures, while the risk factors include, among others, being a women and being at high risk of contracting the coronavirus [69,70]. The statement is reflected in the study by Al-Amad and Hussein [71]: the anxiety level was independent on age, length of professional work and professional category. According to Mahdee et al. [72], anxiety among dentists working in the hospital and clinic was statistically significantly higher than among those admitting only in the clinic

Reviewer 3 Report

It's an interesting paper highlighting the attitude of polish dental professionals during the COVID-19 pandemic. 

An additional information (if possible) regarding the speciality or areas of expertise of the dental professionals might be beneficial.

Additionally, the authors have to cite:

Karayürek F, Yilmaz Çırakoğlu N, Gülses A, Ayna M. Awareness and Knowledge of SARS-CoV-2 Infection among Dental Professionals According to the Turkish National Dental Guidelines. Int J Environ Res Public Health. 2021 Jan 8;18(2):442. doi: 10.3390/ijerph18020442.

and

Karayürek F, Çebi AT, Gülses A, Ayna M. The Impact of COVID-19 Vaccination on Anxiety Levels of Turkish Dental Professionals and Their Attitude in Clinical Care: A Cross-Sectional Study. Int J Environ Res Public Health. 2021 Oct 1;18(19):10373. doi: 10.3390/ijerph181910373.

Best regards

Author Response

Response to Reviewer 3 Comments

Dear Reviewer,

We would like to express our gratitude for your attention concerning our manuscript entitled "Knowledge and attitude of Polish dental healthcare professionals during the COVID-19 pandemic" and the useful comments and suggestions helping us to improve the paper. We truly appreciate the constructive criticisms of you. We acknowledge these realties as limitations of our work. Yours efforts and insights were a tremendous help to us during this revision.

We have considered them carefully and made the required revision as addressed point-by-point below. We hope that you find these revisions an improvement and believe that the revised version can meet yours requirements.

We will appreciate the verification whether the present version of the manuscript fulfils the criteria and can be accepted as sufficiently modified. The authors will welcome further constructive comments if any.

Point 1: An additional information (if possible) regarding the speciality or areas of expertise of the dental professionals might be beneficial.

Response 1: Thank you for your comment. Regarding dentists' specializations – we have added some more information about specializations along with the percentage results in the text under Table 1.

Revised text: Among the respondents, 41.5% had specializations, including general dentistry (58.1%), pediatric dentistry (17.3%), conservative dentistry with endodontics (16.3%), dental surgery ( 9.3%), dental prosthetics (6.6%), orthodontics (6%) and periodontology  (3.7%).

Point 2: Additionally, the authors have to cite:

Karayürek F, Yilmaz Çırakoğlu N, Gülses A, Ayna M. Awareness and Knowledge of SARS-CoV-2 Infection among Dental Professionals According to the Turkish National Dental Guidelines. Int J Environ Res Public Health. 2021 Jan 8;18(2):442. doi: 10.3390/ijerph18020442.

Response 2: Thank you for this valuable suggestion. We have added this paper to the manuscript.

[41] Karayürek F, Yilmaz Çırakoğlu N, Gülses A, Ayna M. Awareness and Knowledge of SARS-CoV-2 Infection among Dental Professionals According to the Turkish National Dental Guidelines. Int J Environ Res Public Health. 2021 Jan 8;18(2):442. doi: 10.3390/ijerph18020442.

Revised text:

  1. In our and other surveys, the respondents correctly selected the basic symptoms of SARS-Cov-2 infection [38,41].
  2. In comparison, in Karayürek et al. study [41 ] 30.4% dentist did not use any antiseptic solutions before dental examination.
  3. An abscess and acute surgical procedures was also one of the mostly approved treatment [41,47].
  4. Almost all respondents (90.7%) used disposable surgical gowns as did other dentists [41,51] and on the contrary to Cagetti et al. study [55] - 21.09%.
  5. Most of respondents admitted concerns about their professional future [12,41,58].

Point 3: Additionally, the authors have to cite:

Karayürek F, Çebi AT, Gülses A, Ayna M. The Impact of COVID-19 Vaccination on Anxiety Levels of Turkish Dental Professionals and Their Attitude in Clinical Care: A Cross-Sectional Study. Int J Environ Res Public Health. 2021 Oct 1;18(19):10373. doi: 10.3390/ijerph181910373.

Response 3: Thank you for another valuable suggestion. We have added this paper to the manuscript.

[65] Karayürek F, Çebi AT, Gülses A, Ayna M. The Impact of COVID-19 Vaccination on Anxiety Levels of Turkish Dental Professionals and Their Attitude in Clinical Care: A Cross-Sectional Study. Int J Environ Res Public Health. 2021 Oct 1;18(19):10373. doi: 10.3390/ijerph181910373.

Revised text:

  1. Actually, the vaccination had a positive effect on reducing the level of anxiety and fear of dental healthcare professionals [65].
  2. However, the use of PPE should remain the main concern regardless of the vaccination [65].

Reviewer 4 Report

This is an interesting article in terms of knowledge and attitude of Polish dental healthcare professionals during the COVID-19 pandemic. However authors put some extensive work into it, an entire manuscript reads like a plagiarism of a very similar, highly cited work published in MDPI IJERPH on January 31th 2021 https://pubmed.ncbi.nlm.nih.gov/33572669/ . Which is even worse, authors did not even bother to incorporate it in Discussion section. This is a major flaw of the study and renders an entire work unpublishable at this stage. Also, according to anti-plagiarism software other similar works (20%) were copy-pasted and merged into this manuscript either, without even simple paraphrasing e.g. L29-41. Some parts need extensive editing as they are directly taken from the source without citing the reference - this is the lowest standard in modern science which is completely unacceptable and needs to be stigmatized. More remarks below:

Introduction

L30 - '(Corona Virus Disease)' is redundant

L38-39 - 'Data on SARS-Cov-2 in relation to dental procedures was not available and supported by scientific papers' - this statement is not true as initial recommendations were available, use rather term 'limited data'

L41-42 ' Dental care in Poland is provided mainly by private dental healthcare professionals. Patients pay dentists directly.' - this statements need relevant citation(s), with some descriptive statistics

L57-61 - hypothesis has been plainly plagiarised from the article I brought up in first few paragraphs of this review and needs to be re-formulated. Also study expectations should be emphasized here.

Materials and methods

L65 - Dental Associations names should be elaborated here

L74-75 - 'The survey was accomplished in collaboration with the Polish Association of Paediatric Dentistry' - this statement is redundant, either clarify or remove it

L102 -  Statistica 13.1 program requires proper manufacturers' data

L103 - Provide name and origin of the Commitee in full details here as well

Results

Tools used for quantification are very poorly described.

There were only p values provided - medians are missing in all comparisons, e.g. Mann-Whitney test. The same applies to graphs. For continuous work in pandemic also tables and graphs should be provided. According to those Table 4 requires major changes.

Statistical methods description should be extended.

Discussion

L186 - Authors start with comparing Polish Dentist with Turkish - this country is quite afar and not even in EC, which results in completely different systemic and state regulations. Likewise, Jordan is being compared shortly after. Answer for for hypothesis should start with relating to similar publications from the same country (Tysiac-Mista et al, Dalewski et al but there were more released). Also there are plenty formidable papers scrutinizing pandemic problems in dental industry published in adjacent European countries e.g. Germany, Finland, Sweden, Denmark, Italy etc. An entire section should be re-written accroding to this remark.

In general, an entire section is too long and reads like a literature review. Authors should write it again according to typical scientific requirements applicable for discussions.

Conclusions

This section is also overlengthy and at least part of it might be moved into Introduction and Discussion section. This part should be kept brief and concise enough to provide answers for a stated hypothesis, no more no less.

Author Response

Response to Reviewer 4 Comments

Dear Reviewer,

We would like to express our gratitude for your attention concerning our manuscript entitled "Knowledge and attitude of Polish dental healthcare professionals during the COVID-19 pandemic" and the useful comments and suggestions helping us to improve the paper. We truly appreciate the constructive criticisms of you. We acknowledge these realties as limitations of our work. Yours efforts and insights were a tremendous help to us during this revision.

We have considered them carefully and made the required revision as addressed point-by-point below. We hope that you find these revisions an improvement and believe that the revised version can meet your requirements.

We will appreciate the verification whether the present version of the manuscript fulfils the criteria and can be accepted as sufficiently modified. The authors will welcome further constructive comments if any.

Point 1: Introduction

L30 - '(Corona Virus Disease)' is redundant

Response 1: Thank you for your comment. We have made the change as recommended.

Revised text: At the end of 2019, the first cases of COVID-19 were diagnosed in Wuhan, China [1,2].

Point 2: Introduction

L38-39 - 'Data on SARS-Cov-2 in relation to dental procedures was not available and supported by scientific papers' - this statement is not true as initial recommendations were available, use rather term 'limited data'

Response 2: Thank you for your comment. We have made changes. As for the explanation (the whole situation is described in detail by Tysiąc-Miśta and Dziedzic in her article): on March 16, 2020 the majority of dental offices in Poland were closed. This was before the first Polish guidelines appeared – by The Polish Dental Association (March 19) and the Ministry of Health (March 25).

Revised text: Data on SARS-Cov-2 in relation to dental procedures was limited and insufficiently supported by scientific papers.

Point 3: Introduction L41-42 - ' Dental care in Poland is provided mainly by private dental healthcare professionals. Patients pay dentists directly.' - this statements need relevant citation(s), with some descriptive statistics

Response 3: Thank you for your comment. We have added appropriate citation based on Supreme Audit Office and The Public Opinion Research Center Foundation, with some descriptive statistics.

Revised text: Dental care in Poland is provided mainly by private dental healthcare professionals. Patients pay dentists directly [5]. According to Public Opinion Research Center CBOS  data, the vast majority (69%) of Poles used the services of a private dental clinic (hence 31% of Poles used the services reimbursed by the National Health Fund, i.e. less than 1/3) [6].

Point 4: Introduction L57-61 - hypothesis has been plainly plagiarised from the article I brought up in first few paragraphs of this review and needs to be re-formulated. Also study expectations should be emphasized here.

Response 4: Thank you for your comment. As for the explanation, we did not know the Dalewski’ et al. (2021) paper before writing our manuscript, therefore we did not use it in our sources. To discuss with relating to similar publications from the same country we have added this paper.

Hypothesis and study expectations has been re-formulated.

Revised text: The aim of the study was to investigate knowledge and attitude of Polish dental healthcare professionals during the relatively early stage of COVID-19 pandemic with particular emphasis on the main problems in their work.  The authors expected the current study to (i) provide valuable comparisons on managing an office in a pandemic situation; (ii) gather information about attitudes and approaches to oral health care and protective measures, and (iii) define strategies to facilitate organization of the dental care during the current and likely future situations, as to better confront similar outbreaks.

Point 5: Materials and methods L65 - Dental Associations names should be elaborated here.

Response 5: Thank you for your comment. We have added Dental Associations names.

Revised text: The link to the online survey was sent through an anonymous mailing list to all dentists registered in The Polish Dental Association and Polish Society of Pediatric Dentistry.

Point 6: Materials and methods L74-75 - 'The survey was accomplished in collaboration with the Polish Association of Paediatric Dentistry' - this statement is redundant, either clarify or remove it

Response 6: Thank you for your comment. This statement has been removed.

Point 7: Materials and methods L102 - Statistica 13.1 program requires proper manufacturers' data

Response 7: Thank you for your comment. An information was added – Dell Inc.

Revised text: The statistical analysis was performed using the Statistica 13.1 (Dell Inc.) program.

Point 8: Materials and methods L103 - Provide name and origin of the Commitee in full details here as well

Response 8: Thank you for your comment. We have added name and origin of the Committee (Bioethics Committee of Medical University of Warsaw) to manuscript.

Revised text: The study was approved by Bioethics Committee of Medical University of Warsaw (decision number AKBE101/2020).

Point 9: Results - Tools used for quantification are very poorly described.

Response 9: Thank you for your comment. We have added additional information.

Revised text: The obtained data were stored on a password protected computer drive and. In first step the data were analysed using basic descriptive statistics. Next, the significance of the relationship between the answers and the socio- demographic data was assessed using the Chi square test. However, for age and length of work, the dependence on the socio- demographic data was evaluated using the Mann-Whitney test. P < 0.05 was considered significant in all statistical analyses.

Point 110: Results - There were only p values provided - medians are missing in all comparisons, e.g. Mann-Whitney test.

Response 10: Thank you for this valuable suggestion. We have added additional information.

The modified table is as follows:      `

Table 7. Relationships between age, length of work and selected parameters, evaluated using the Mann-Whitney test.

Category

Age

Length of work

Median

p value for the Mann-Whitney test

Median

p value for the Mann-Whitney test

Keeping up to date with relevant information about COVID-19

(no)

35.40

p=0.0123

3.00

p=0.0002

(occasionally)

34.00

3.00

(yes)

48.00

5.00

Following the guidelines of the Ministry of Health

(no)

45.00

p=0.1819

4.00

p=0.6544

(yes)

43.00

4.30

Following the guidelines of the Polish Dental Society

(no)

40.00

p=0.0167

4.50

p=0.0892

(yes)

45.00

4.00

Are you concerned about the situation related to the COVID-19?

(no)

38.00

p=0.0002

4.00

p=0.0001

(rather no)

36.00

3.00

(rather yes)

44.00

4.00

(very much)

49.00

5.00

Assessment of the risk of infection with SARS-Cov-2 virus at work

(low risk)

41.00

p=0.1202

4.00

p=0.4901

(medium risk)

41.00

4.00

(high risk)

43.00

4.00

(very high risk)

46.00

5.00

(yes)

39.00

4.00

Point 11: Results - The same applies to graphs.

Response 11: Thank you for this valuable suggestion. We have added a new figure using medians in comparisons.

Figure 1. Relationships between usage of a rubber dam during treatment and age (a), length of work (b), the Mann-Whitney test(* statistically significant p<0.005).

Point 12: Results - For continuous work in pandemic also tables and graphs should be provided. According to those Table 4 requires major changes.

Response 12: Thank you for this valuable suggestion. We have prepared additional Table 6 and made changes in Table 7.

The modified tables are as follows:  

Table 6. Relationships between continuing work during a pandemic and selected parameters, evaluated using the chi square test.

Keeping up to date with relevant information about COVID-19

Following the guidelines of the Ministry of Health

Following the guidelines of the Polish Dental Society

Are you concerned about the situation related to the COVID-19?

Assessment of the risk of infection with SARS-Cov-2 virus at work

Continuing work during a pandemic

p=0.0963

p=0.3321

p=0.0709

p=0.0013*

p=0.0106

*Statistically significant (p<0.005)

Table 7. Relationships between age, length of work and selected parameters, evaluated using the Mann-Whitney test.

Category

Age

Length of work

Median

p value for the Mann-Whitney test

Median

p value for the Mann-Whitney test

Keeping up to date with relevant information about COVID-19

(no)

35.40

p=0.0123

3.00

p=0.0002*

(occasionally)

34.00

3.00

(yes)

48.00

5.00

Following the guidelines of the Ministry of Health

(no)

45.00

p=0.1819

4.00

p=0.6544

(yes)

43.00

4.30

Following the guidelines of the Polish Dental Society

(no)

40.00

p=0.0167

4.50

p=0.0892

(yes)

45.00

4.00

Are you concerned about the situation related to the COVID-19?

(no)

38.00

p=0.0002*

4.00

p=0.0001*

(rather no)

36.00

3.00

(rather yes)

44.00

4.00

(very much)

49.00

5.00

Assessment of the risk of infection with SARS-Cov-2 virus at work

(low risk)

41.00

p=0.1202

4.00

p=0.4901

Point 13: Results - Statistical methods description should be extended.

Response 13: Thank you for this valuable suggestion. We have added additional information.

Revised text: The obtained data were stored on a password protected computer drive and. In first step the data were analysed using basic descriptive statistics. Next, the significance of the relationship between the answers and the socio- demographic data was assessed using the Chi square test. However, for age and length of work, the dependence on the socio- demographic data was evaluated using the Mann-Whitney test. P < 0.05 was considered significant in all statistical analyses.

Point 14: Discussion L186 - Authors start with comparing Polish Dentist with Turkish - this country is quite afar and not even in EC, which results in completely different systemic and state regulations.

Response 14: Thank you for this valuable suggestion. As for the explanation, as COVID-19 affect the dental community worldwide, some authors conducted a cross-sectional study using a multisite survey to examine dentists’ knowledge, attitudes toward COVID-19 and the effect on their livelihood [e.g. Bakaeen et al.Dentists' knowledge, attitudes, and professional behavior toward the COVID-19 pandemic: A multisite survey of dentists' perspectives. J Am Dent Assoc. 2021 Jan;152(1):16-24. doi: 10.1016/j.adaj.2020.09.022.]  Based on such examples we have compared different countries, worldwide.

We appreciate your comment and to compare results we have included publications from our country (e.g. Tysiąc-Miśta et al, Dalewski et al., Sycinska-Dziarnowska et al., Nijakowski et al.).

At the same time, we feel confused as another Reviewer required, that “the authors have to cite additional two Turkish papers to this manuscript”:

1.Karayürek, F.; Yilmaz, Çırakoğlu N.; Gülses, A.; Ayna, M. Awareness and Knowledge of SARS-CoV-2 Infection among Dental Professionals According to the Turkish National Dental Guidelines. Int J Environ Res Public Health. 2021 8;18(2):442. doi: 10.3390/ijerph18020442

  1. Karayürek, F.; Çebi, AT.; Gülses, A.; Ayna, M. The Impact of COVID-19 Vaccination on Anxiety Levels of Turkish Dental Professionals and Their Attitude in Clinical Care: A Cross-Sectional Study. Int J Environ Res Public Health. 2021 1;18(19):10373. doi: 10.3390/ijerph181910373.

Revised text: Please see revised Discussion.

Point 15: Discussion - Likewise, Jordan is being compared shortly after. Answer for hypothesis should start with relating to similar publications from the same country (Tysiac-Mista et al, Dalewski et al but there were more released).

Response 15: Thank you for this valuable suggestion. With relating to similar publications from the same country we have added another Polish authorship papers to compare results and, what is more, excluded Jordanian article from our manuscript:

  1. Dalewski, B.; Palka, L.; Kiczmer, P.; Sobolewska, E. The Impact of SARS-CoV-2 Outbreak on the Polish Dental Community’s Standards of Care—A Six-Month Retrospective Survey-Based Study. Int. J. Environ. Res. Public Health 2021, 18, 1281. https://doi.org/10.3390/ ijerph18031281.
  2. Sycinska-Dziarnowska M, Paradowska-Stankiewicz I. Dental Challenges and the Needs of the Population during the Covid-19 Pandemic Period. Real-Time Surveillance Using Google Trends. Int J Environ Res Public Health. 2020;17(23):8999. Published 2020 Dec 3. doi:10.3390/ijerph17238999.
  3. Nijakowski K, Cieślik K, Łaganowski K, Gruszczyński D, Surdacka A. The Impact of the COVID-19 Pandemic on the Spectrum of Performed Dental Procedures. Int J Environ Res Public Health. 2021 Mar 25;18(7):3421. doi: 10.3390/ijerph18073421.

Revised text: Please see revised Discussion section.

Point 16: Discussion - Also there are plenty formidable papers scrutinizing pandemic problems in dental industry published in adjacent European countries e.g. Germany, Finland, Sweden, Denmark, Italy etc. An entire section should be re-written according to this remark.

Response 16:  Thank you for this valuable suggestion. An entire section has been re-written according to your remark.

In our manuscript there can be found papers comparing dental pandemic situation in adjacent countries: Poland, Italy (several articles), UK, Switzerland and Liechtenstein. Due to your valuable suggestion, we added papers scrutinizing pandemic problems in Czech Republic, Russia, Central Europe (Belgium, Austria, Germany, Switzerland) as follows:

  1. Schwendicke F, Krois J, Gomez J. Impact of SARS-CoV2 (Covid-19) on dental practices: Economic analysis. J Dent. 2020;99:103387. doi:10.1016/j.jdent.2020.103387
  2. Wolf, T. G., Deschner, J., Schrader, H., Bührens, P., Kaps-Richter, G., Cagetti, M. G., & Campus, G. (2021). Dental Workload Reduction during First SARS-CoV-2/COVID-19 Lockdown in Germany: A Cross-Sectional Survey. International journal of environmental research and public health18(6), 3164. https://doi.org/10.3390/ijerph18063164
  3. Wiesmüller, V., Bruckmoser, E., Kapferer-Seebacher, I., Fink, K., Neururer, S., Schnabl, D., & Laimer, J. (2021). Dentists' Working Conditions during the First COVID-19 Pandemic Lockdown: An Online Survey. Healthcare (Basel, Switzerland)9(3), 364. https://doi.org/10.3390/healthcare9030364
  4. Mekhemar, M.; Attia, S.; Dörfer, C.; Conrad, J. The Psychological Impact of the COVID-19 Pandemic on Dentists in Germany. J Clin Med. 2021, 10, 5, 1008. doi:10.3390/jcm10051008
  5. Schmidt J, Waldova E, Balkova S, Suchanek J, Smucler R. Impact of COVID-19 on Czech Dentistry: A Nationwide Cross-Sectional Preliminary Study among Dentists in the Czech Republic. Int J Environ Res Public Health. 2021;18(17):9121. Published 2021 Aug 29. doi:10.3390/ijerph18179121
  6. Carvalho, J.C.; Declerck, D.; Jacquet, W.; Bottenberg, P. Dentist Related Factors Associated with Implementation of COVID-19 Protective Measures: A National Survey. Int. J. Environ. Res. Public Health 2021, 18, 8381. https://doi.org/10.3390/ijerph18168381
  7. Sarapultseva, M.; Zolotareva, A.; Kritsky, I.; Nasretdinova, N.; Sarapultsev, A. Psychological Distress and Post-Traumatic Symptomatology among Dental Healthcare Workers in Russia: Results of a Pilot Study. Int. J. Environ. Res. Public Health 2021, 18, 708. https://doi.org/10.3390/ijerph18020708

Point 17: Discussion - In general, an entire section is too long and reads like a literature review. Authors should write it again according to typical scientific requirements applicable for discussions.

Response 17: Thank you for this valuable suggestion. An entire section has been re-written according to your comment. The part of Discussion has been moved into Introduction. We hope you find it improved.

Revised text: Please see revised Discussion section.

Point 18: Conclusions - This section is also overlengthy and at least part of it might be moved into Introduction and Discussion section. This part should be kept brief and concise enough to provide answers for a stated hypothesis, no more no less.

Response 18: Thank you for this valuable suggestion. Some parts of Conclusions have been removed. Conclusions section has been trimmed according to your comment to kept brief and concise enough. We hope you find it improved.

Revised text: In summary, the current study focused on the early stage (first wave) of the pandemic.  The study provides valuable comparisons on managing an office in a pandemic situation; gather information about attitudes and approaches to oral health care and protective measures, and define strategies to facilitate organization of the dental care during the current and likely future situations, as to better confront similar outbreaks. Most of the respondents follow the information about the COVID-19 pandemic on an ongoing basis and have knowledge about COVID-19 infection. Respondents are concerned and anxious about the situation. During that time, the respondents see patients with abscesses and tooth injuries; without sandblasting, rarely performing scaling and permanent prosthetic restorations. Most respondents take preventive measures against the virus, both for patients and themselves. At the time of the study, almost half plan to vaccinate.  Continuing work during a pandemic, age and length of work is significantly related to being concerned about the COVID-19 situation.  Age and length of work is significantly related to usage of a rubber dam during treatment and keeping up to date with relevant information. Overall, this study gives insight into the reactions of this professional group during the early phase of one of the most important public health challenges in recent years.

Round 2

Reviewer 4 Report

An entire manuscript improved into the publishable form eventually. Good work!